# Targeting the p53/xCT/GSH Axis with PRIMA-1^Met^ Combined with Sulfasalazine Shows Therapeutic Potential in Chronic Lymphocytic Leukemia

**DOI:** 10.3390/ijms26125559

**Published:** 2025-06-10

**Authors:** Martina Pasino, Andrea Speciale, Silvia Ravera, Giovanna Cutrona, Rosanna Massara, Nadia Bertola, Maurizio Viale, Irena Velkova, Andrea Nicola Mazzarello, Franco Fais, Fabrizio Loiacono, Serena Matis, Giulia Elda Valenti, Nicola Traverso, Cinzia Domenicotti, Barbara Marengo, Bruno Tasso, Adalberto Ibatici, Emanuele Angelucci, Tiziana Vaisitti, Paola Monti, Paola Menichini

**Affiliations:** 1Neuro-Oncology and Mutagenesis Unit, IRCCS Ospedale Policlinico San Martino, 16132 Genoa, Italy; martina.pasino01@gmail.com (M.P.); andrea.speciale@galliera.it3 (A.S.); irena.velkova@hsanmartino.it (I.V.); paola.monti@hsanmartino.it (P.M.); 2Department of Experimental Medicine, University of Genoa, 16132 Genoa, Italy; silvia.ravera@unige.it (S.R.); andreanicola.mazzarello@edu.unige.it (A.N.M.); franco.fais@unige.it (F.F.); giuliaelda.valenti@edu.unige.it (G.E.V.); nicola.traverso@unige.it (N.T.); cinzia.domenicotti@unige.it (C.D.); barbara.marengo@unige.it (B.M.); 3IRCCS Ospedale Policlinico San Martino, 16132 Genoa, Italy; 4Molecular Pathology Unit, IRCCS Ospedale Policlinico San Martino, 16132 Genoa, Italy; giovanna.cutrona@hsanmartino.it (G.C.); rosanna.massara@hsanmartino.it (R.M.); nadia.bertola@hsanmartino.it (N.B.); 5Biotherapy Unit, IRCCS Ospedale Policlinico San Martino, 16132 Genoa, Italy; maurizio.viale@hsanmartino.it; 6Cytometry Facility, IRCCS Ospedale Policlinico San Martino, 16132 Genoa, Italy; fabrizio.loiacono@hsanmartino.it; 7Molecular Oncology and Angiogenesis Unit, IRCCS Ospedale Policlinico San Martino, 16132 Genoa, Italy; serena.matis@hsanmartino.it; 8Department of Pharmacy, University of Genoa, 16132 Genoa, Italy; bruno.tasso@unige.it; 9Hematology and Cellular Therapy Unit, IRCCS Ospedale Policlinico San Martino, 16132 Genoa, Italy; adalberto.ibatici@hsanmartino.it (A.I.); emanuele.angelucci@hsanmartino.it (E.A.); 10Department of Medical Sciences, University of Torino, 10124 Turin, Italy; tiziana.vaisitti@unito.it

**Keywords:** P53, PRIMA-1^Met^, GSH, SLC7A11/xCT, antioxidant defenses, oxidative stress

## Abstract

In Chronic Lymphocytic Leukemia (CLL), mutations at the *TP53* tumor suppressor gene are an important hallmark since they may strongly influence the therapeutic decision. PRIMA-1^Met^ (also known as APR-246/Eprenetapopt) is a small molecule able to restore the wild-type (wt) p53 conformation to mutant p53 proteins and to stimulate apoptosis in tumor cells; in addition, it can deplete the glutathione reservoir, increasing reactive oxygen species (ROS) production. In this study, we investigated whether combining PRIMA-1^Met^ with Sulfasalazine (SAS), a SLC7A11/xCT inhibitor, reduces CLL cell viability by targeting mutant p53 and the glutathione pathway. The results demonstrated that, in CLL cells, PRIMA-1^Met^ did not restore the wt functions in the mutant p53 proteins, but it strongly reduced the antioxidant defense and induced cell death. PRIMA-1^Met^ and SAS combination synergistically reduced cell survival regardless of p53 status and further impaired antioxidant capacity, especially in mutant p53 cells, linking their cytotoxic effect to redox imbalance. Thus, the association of PRIMA-1^Met^ with drugs targeting the antioxidant response could represent a valid strategy to kill CLL cells carrying either wt or mutant p53.

## 1. Introduction

Chronic Lymphocytic Leukemia (CLL) is a highly heterogeneous form of leukemia with an equally heterogeneous clinical course, ranging from rapid disease progression to decades of survival without treatment [1,2]. At diagnosis CLL is characterized by complex karyotype aberrations, the most frequent being partial deletions at 13q (~55%), 11q (~15%), 17p (~8%), gain of chromosome 12 (~15%), and by mutations at different genes including *TP53*, *SF3B1*, *BIRC3*, *NOTCH1*, and *ATM* [3,4].

Mutations in the *TP53* gene represent a critical hallmark of the disease, as they can significantly impact the success of treatment protocols. A low incidence of *TP53* mutations, ranging from 5 to 7%, is found at diagnosis, while it rises to approximately 40% in refractory CLL [5,6,7]. For this reason, the disruption of the *TP53* gene that may occur by deletion at chromosome 17p13.1 (del17p) and/or by mutations is considered the most important predictive biomarker in CLL [8,9]. Indeed, the European Research Initiative on CLL (ERIC) group recommends *TP53* mutational screening for all patients before the start of therapy to avoid a treatment that may be ineffective for patients carrying *TP53* mutations [10].

The *TP53* tumor suppressor gene encodes a tetrameric transcription factor that controls a plethora of key pathways such as cell cycle regulation, DNA damage response, apoptosis, senescence, DNA repair, cell migration, and autophagy, all determinants for cell homeostasis maintenance [11,12]. More recently, aerobic energy metabolism has also been added to the long list of functions modulated by this key protein [13,14,15]. Importantly, *TP53* alterations found in cancer, as in the case of CLL, not only may abolish the transactivating activity of the protein toward target genes, but, in many cases, may promote the onset of new deleterious functions, defined as Gain of Functions, which may strongly affect cancer progression and the response to therapy.

Nowadays, several new treatments are available for CLL patients carrying *TP53* alterations. In particular, ibrutinib and idelalisib, which inhibit the signaling pathway initiated by the B-cell antigen receptor, or venetoclax, which facilitates cell apoptosis, have contributed to overcoming the low efficacy of chemoimmunotherapy in patients with del(17p) and/or *TP53* mutations [16,17]. However, despite the development of these new drugs, *TP53* alterations may be associated with therapy resistance and a shorter progression-free survival [16], making it necessary to develop alternative targeted therapies capable of improving the success of treatments in patients carrying *TP53* mutations.

Several molecules able to target mutant p53 for reactivation have been isolated in the last years. PRIMA-1^Met^/APR-246/Eprenetapopt (hereafter defined as PRIMA-1^Met^) is the most clinically advanced agent targeting mutant p53, which has been shown to reactivate the pro-apoptotic functions of wt p53 in the mutated forms of p53, eventually exerting potent anti-tumor activity in preclinical models [18]. Although the activity of PRIMA-1^Met^ has been largely attributed to its capacity to reactivate the wt functions of a mutated p53 protein, promoting apoptotic cell death, it can exert additional effects by antagonizing glutathione and thioredoxin reductase systems, leading to reactive oxygen species (ROS) overproduction [19,20,21,22]. The involvement of mutant p53 proteins in the redox effects of PRIMA-1^Met^ has been addressed by Liu et al. [23,24], reporting that mutant p53 sensitized tumor cells to PRIMA-1^Met^-induced oxidative stress, inhibiting glutathione synthesis through the inhibition of the xC-system. In particular, they found that the binding of mutant p53 to NRF2 can suppress the transcription of SCL7A11, a key component of the xC- system, reducing the glutathione content and increasing intracellular ROS; specifically, SLC7A11 is responsible of the importing of cystine, a dimer of cysteine that represents the limiting amino acid for glutathione de novo synthesis. In the presence of PRIMA-1^Met^, the glutathione reservoir is further depleted by direct binding of the PRIMA-1^Met^-derived active compound to glutathione cysteines, inactivating the antioxidant properties of GSH and promoting further ROS production and cell death. Therefore, the expression of SLC7A11 has been proposed as a predictive biomarker of PRIMA-1^Met^ sensitivity [23,24]. Interestingly, the pharmacological inhibition of SLC7A11 can give rise to a synthetic lethal interaction with the mutant p53, highly expressed in tumor cells, resulting in high oxidative stress and massive cell death [25].

The xC-transporter may be inhibited by Sulfasalazine (SAS), an anti-inflammatory drug belonging to the class of sulphonamides, used for the treatment of chronic inflammatory bowel diseases and rheumatoid arthritis [26] (Wahl et al., 1998). SAS is known to inhibit the Nuclear Factor κB (NF-kB), which is involved in pro-inflammatory pathways [26], and is constitutively activated in CLL patients [27]. The p53-independent inhibition of the xC system by SAS results in decreased levels of glutathione. Thus, SAS employment in association with PRIMA-1^Met^ could further increase the intracellular oxidative stress and sensitize CLL cells carrying mutant p53 or expressing a wt p53 but experiencing a therapy-resistant phenotype.

The cytotoxic potential of PRIMA-1^Met^ has been recently investigated in 62 clinical CLL samples characterized for *TP53* mutations and p53 protein levels [28]. A comparable viability reduction and apoptosis induction following PRIMA-1^Met^ were observed in *TP53*-mutated and *TP53* wt samples. Nevertheless, in a subset of samples accumulating mutant p53, PRIMA-1^Met^ induced a reduction of mutant p53 protein level that correlated with a decrease in cell viability compared to that of cells expressing stable levels of mutant p53 protein. Therefore, the amount of p53 mutant protein was reduced in those CLL cells that were sensitive to PRIMA-1^Met^ [28].

Besides this study on PRIMA-1^Met^ effects in CLL, to date, no investigations have been conducted to explore the activity of PRIMA-1^Met^ in combination with drugs involved in the modulation of the antioxidant response in CLL cells. In the present paper, we investigate the effect of PRIMA-1^Met^ alone or in combination with SAS in cells from CLL patients and CLL cell lines carrying a wt or a mutant p53 protein. Our findings indicate that in CLL cells, the combination of PRIMA-1^Met^ with SAS significantly decreases antioxidant capacity and increases ROS production, leading to a synergistic reduction in cell survival in both wt and mutant p53 carrying cells.

## 2. Results

### 2.1. PRIMA1^Met^ and SAS Combined Treatment Induces Apoptosis in CLL Cell Lines

To investigate the impact of PRIMA-1^Met^ in combination with SAS on CLL cell survival, we used the OSU and MEC-1 CLL cell lines, carrying wt and mutant p53 proteins, respectively. The TP53 mutation expressed in MEC-1 cells produces a truncated p53 protein, resulting in a faster migrating band in Western blot analysis (Figure 1A).

Notably, compared to OSU, MEC-1 expressed lower levels of SLC7A11/xCT (Figure 1A,B) suggesting a potential sensitivity to PRIMA1^Met^ treatment; indeed, it has been demonstrated that the expression of SLC7A11 may represent a superior determinant of response to PRIMA-1^Met^ compare to the mutational status of p53 alone in cancer cells [24]. On the other hand, MEC-1 cells express a slightly higher level of Bcl-2 (Figure 1A,B), hypothesizing an intrinsic anti-apoptotic signature as already reported in the literature [29].

Apoptosis induction was determined by the Annexin V/PI assay after 24 h of treatment with PRIMA-1^Met^ (5 μM and 10 μM) or SAS (300 μM) alone and in combination. A higher percentage of apoptotic cells for untreated OSU than for MEC-1 cells (20% and 9%, respectively) was found (Figure 2A), in keeping with the lower Bcl-2 level observed in former cells (Figure 1). PRIMA-1^Met^ (5 μM and 10 μM) or SAS (300 μM) alone were not able to induce apoptosis in either cell line at these concentrations, while the co-treatments increased the apoptotic population in a dose-dependent manner: 5 μM PRIMA-1^Met^ + 300 μM SAS induced a 30% of apoptosis in OSU cells and 17.7% in MEC-1 cells (approximately 1.5- and 2-fold compared to untreated cells, respectively); 10 μM PRIMA-1^Met^ + 300 μM SAS induced the 49% of apoptosis in OSU cells and 37% in MEC-1 cells (approximately 2.5- and 4-fold compared to untreated cells, respectively).

The percentage of apoptotic cells was always lower in MEC-1 cells than in OSU, confirming an apoptotic-resistant behaviour. This response is accompanied by the trend of Bcl-2 that, while decreasing after PRIMA-1^Met^ and SAS treatments in OSU cells, slightly increased in MEC-1, except at the highest 10 μM concentration of PRIMA-1^Met^ in combination with SAS (Figure 2B,C). Despite this, the MEC-1 apoptotic population increased by approximately 4-fold compared to untreated cells; thus, the co-treatment of PRIMA-1^Met^ and SAS was able to significantly induce cell death also in MEC-1 cells, counteracting their apoptosis resistance.

### 2.2. PRIMA1^Met^ and SAS Synergistically Decrease Survival in CLL Cell Lines, Regardless of Their p53 Status

Cell viability was also assayed by MTT analysis at 24 h and 48 h of treatment (Figure 3). The 5 μM and 10 μM PRIMA-1^Met^ alone or 300 μM SAS did not significantly modify the percentage of cell survival, while 20 μM PRIMA-1^Met^ did reduce OSU cell viability after 48 h treatments (Figure 3A) and MEC-1 cell viability after 24 h and 48 h of treatment (Figure 3B).

Concerning combined treatment, the percentage of survival after 5 μM PRIMA-1^Met^ plus SAS, compared to untreated cells, was 90.7% in OSU and 51.2% in MEC-1 after 24 h. After 48 h of treatment, it was 33.7% and 24.2% in OSU and MEC-1 cells, respectively. The combination of 10 μM PRIMA1^Met^ plus SAS induced an analogous effect after 24 h treatment, while no surviving cells were detected after 48 h. Notably, this assay, based on the metabolic activity of the cells, revealed that MEC-1 cells appeared slightly more sensitive than OSU cells.

Additional MTT experiments were conducted to define the synergistic behavior of the two compounds (Figure 3C,D). When OSU and MEC-1 cells were treated with combinations of different concentrations of PRIMA-1^Met^ and SAS, we observed that PRIMA-1^Met^ synergized with SAS at the lowest drug concentrations (5μM and 10 μM PRIMA-1^Met^), up to 1200 μM SAS, giving a D value <1. The concentration of 20 μM PRIMA-1^Met^ can be considered additive (D = 1) only when higher SAS concentrations were used, while at 40 μM PRIMA-1^Met^, the activity of the combination of the two drugs is never synergistic (D > 1).

Altogether, these data indicate that the combination of PRIMA-1^Met^ and SAS can synergistically influence CLL cell survival, irrespective of their p53 status.

### 2.3. PRIMA-1^Met^ and SAS Treatment Reduce the Survival of Primary CLL Cells Carrying Wild-Type and Mutant p53 Protein

Primary CLL cells from patients carrying wt and mutant p53 proteins (see Appendix A) were also assayed for PRIMA-1^Met^ and SAS sensitivity (Figure 4). The molecular features of the samples are outlined in Appendix A. Cells were exposed to PRIMA-1^Met^ (as indicated in the figure legend) in the presence of CpG-ODN/IL-15 as an activating and microenvironment-mimicking stimulus [30]. The percentage of viable stimulated cells at 24 h, 48 h, and 72 h following PRIMA-1^Met^ alone, or in combination with 300 μM SAS, is reported in Figure 4 for wt (Figure 4A) and mutant p53 (Figure 4B) carrying cells.

PRIMA-1^Met^ exposure resulted in a significant concentration- and time-dependent viability decrease in all CLL samples. The treatment with SAS was not toxic per se; when administered with PRIMA-1^Met^ it slightly reduced cell viability in comparison to treatment with PRIMA-1^Met^ alone. It has to be noted that with this assay, the response of primary CLL samples to PRIMA-1^Met^/SAS treatment appeared heterogeneous, ranging from cells strongly responding to the addition of SAS to those slightly or not responding (Appendix A). However, a preliminary MTT experiment conducted on primary CLL cells with a broader range of drug concentrations revealed a synergistic activity of PRIMA-1^Met^ and SAS, particularly at low drug doses (Appendix A), indicating that the PRIMA-1^Met^ and SAS combination exerts a synergic effect on CLL patients’ cells.

A comparable percentage of survival following the treatments was observed in quiescent non-activated cells (Appendix A), indicating that the decrease in cell viability in response to PRIMA-1^Met^ and SAS was not restricted to CpG-ODN/IL-15 stimulated cells. Remarkably, PRIMA-1^Met^ was not cytotoxic for peripheral blood mononuclear cells (PBMC) from healthy donors treated with the PRIMA-1^Met^ dose range used for CLL primary cells (Appendix A); indeed, in PBMC, the IC_50_ was reached at considerably higher PRIMA-1^Met^ concentrations than in CLL primary cells (Appendix A). Thus, low PRIMA-1^Met^ concentrations can be considered very cytotoxic in primary CLL cells, regardless of their p53 status.

### 2.4. PRIMA-1^Met^ Does Not Restore the Transcriptional Activity of Mutant p53 in CLL Cells

As shown in many tumor cells, PRIMA-1^Met^ may reactivate the wt functions of mutant p53, allowing it to convert the mutated protein into a transcriptionally active form, able to activate target genes [31]. To investigate the modulation of wt and mutant p53, with their relative targets, following treatments with PRIMA-1^Met^, Western blots were performed, and the levels of p53, MDM2, and p21 were determined in OSU and MEC-1 cells. MEC-1 cells express a truncated form of p53 instead of an unfolded full-length protein; thus, this mutant p53 protein is likely unable to assume a wt-like conformation following PRIMA-1^Met^ treatment. However, these analyses could reveal other PRIMA-1^Met^ effects on p53 modulation unrelated to p53 protein reactivation as a transcription factor. In OSU cells, we observed a p53 protein increase following 10 μM and 20 μM PRIMA-1^Met^ (Figure 5A), while in MEC-1 cells, we observed a p53 increase at 5 μM PRIMA-1^Met^, and then a dose-dependent reduction of mutant p53 (Figure 5B). In both cell lines, MDM2 and p21 proteins followed the same trend of p53 expression, with an overall protein reduction in 20 μM PRIMA–1^Met^-treated MEC-1 cells. Thus, in this cell context, no reactivation of mutant p53 by PRIMA-1^Met^ could be detected.

The modulation of wt and mutant p53, as well as MDM2 and p21, following PRIMA-1^Met^ exposure was also evaluated in primary CLL cell samples (Figure 6).

In cells expressing wt p53, we observed an increase in p53 level that correlated with an increase in MDM2 and p21 (Figure 6A). The p53 expression was also evaluated in the presence of CpG-ODN/IL15 pre-treatment (Figure 6B); the 24 h pre-treatment with CpG-ODN/IL15 boosted this induction already in control cells, and, consequently, also MDM2 and p21 appeared strongly induced (Figure 6B). In contrast, in cells carrying mutant p53, we observed a decrease in mutant p53 level as PRIMA-1^Met^ concentration increased, and no signal for p21 and MDM2 was detected (Figure 6C). After a 24 h pre-treatment with CpG-ODN/IL15, the level of mutant p53 and p21 increased at the 5 μM PRIMA-1^Met^ concentration but decreased afterward (Figure 6D). However, no MDM2 induction was detected, suggesting the lack of mutant p53 reactivation by PRIMA-1^Met^ in these experimental conditions.

### 2.5. PRIMA-1^Met^ and Sulfasalazine Significantly Induce SLC7A11/xCT in MEC-1 CLL Cell Line

Several studies have shown that PRIMA-1^Met^ treatment leads to an increase in SLC7A11/xCT protein, known to promote glutathione synthesis, contributing to the maintenance of the redox balance in tumor cells carrying mutant p53 [23,32,33]. Thus, the level of SLC7A11/xCT was analyzed in CLL cells after 24 h of treatment with PRIMA-1^Met^. In addition, since SAS is known to inhibit SLC7A11/xCT, the level of SLC7A11/xCT following the combined PRIMA-1^Met^ and SAS was also determined (Figure 7).

In OSU cells, a slight, not significant increase in SLC7A11/xCT after PRIMA-1^Met^ alone was observed, while SAS and the combined PRIMA-1^Met^/SAS treatment induced a significant expression of SLC7A11/xCT compared to untreated cells. In MEC-1, an increase in SLC7A11/xCT was already detectable following 20 μM PRIMA-1^Met^ and further raised after co-treatment with SAS (Figure 7A,B). Interestingly, at 5 µM PRIMA-1^Met^/SAS combination, the level of SLC7A11/xCT was higher in MEC-1 cells than in OSU cells, suggesting a stronger susceptibility of CLL cells expressing mutant p53 to an oxidative stress condition. The SLC7A11/xCT protein induction, particularly evident following SAS exposure, may derive from the increased cellular oxidative stress caused by the molecule itself (in virtue of its ability to deplete intracellular GSH) which, in turn, triggers the synthesis of further SLC7A11/xCT to allow cells to cope with a pro-oxidative environment. Indeed, as shown in Appendix A, the treatment with H_2_O_2,_ administered as a positive control of oxidative stress, induced the expression of SLC7A11/xCT, while the treatment with N-acetylcysteine, a synthetic precursor of intracellular cysteine for GSH synthesis, alone or in combination with SAS, was able to prevent the redox-dependent induction of SLC7A11/xCT expression.

### 2.6. PRIMA1^Met^ and SAS Deplete the Intracellular Reduced Glutathione (GSH) Content and Enhance ROS Levels Mainly in MEC-1 Cells

As widely reported, a relevant activity of PRIMA-1^Met^ is the modulation of the cellular redox system [34]. PRIMA-1^Met^, by a direct action on GSH cysteines, has been shown to reduce the level of glutathione itself, increasing intracellular ROS and contributing to its anticancer activity [21,23,32]. To better clarify the role of pro-oxidant conditions in the PRIMA-1^Met^-induced SLC7A11/xCT increase, we measured the levels of reduced glutathione (GSH), the active form of glutathione involved in protection against oxidative stress, and ROS following PRIMA-1^Met^ alone and in combination with SAS.

By using an HPLC assay, we found that untreated OSU cells presented a higher level of GSH than MEC-1 cells (Figure 8A) in keeping with the high level of SLC7A11/xCT protein determined in OSU cells (Figure 1A,B). Instead, 5 μM PRIMA1^Met^ induced a GSH depletion of about 50% in OSU and 30% in MEC-1 cell lines, while 10 μM PRIMA-1^Met^ led to a GSH depletion of 23% and 55% compared to untreated cells in OSU and MEC-1 cells, respectively.

The treatment with SAS alone induced a depletion of 80% in OSU and 90% in MEC-1 cells, while a nearly complete depletion of GSH was measured after co-treatments in both cell lines. These results show that PRIMA-1^Met^ per se induced a GSH decrease that was stronger in MEC-1 than in OSU cells, while a massive GSH reduction in both cell lines was then elicited by SAS.

The depletion of GSH was not associated with a significant increase in ROS in OSU cells after treatment with PRIMA1^Met^ alone, while it was associated with an increasing trend of ROS in MEC-1 cells (Figure 8B). The co-treatments induced a slight increase in ROS compared to single treatment in OSU, whereas elevated levels of ROS were observed in MEC-1 cells, demonstrating the pro-oxidative effect of these molecules in a mutant p53 environment and indicating that MEC-1 cells are more sensitive to GSH depletion compared to OSU. Thus, MEC-1 cells showed an impaired ability to manage such a remarkable GSH reduction and, consequently, the tendency to accumulate a higher amount of ROS.

### 2.7. PRIMA 1^Met^ and SAS Significantly Affect the Balance Between Oxidative Stress Production and Antioxidant Defenses in OSU and MEC-1 Cells

To assess whether treatment with PRIMA-1^Met^ and SAS, alone or in combination, modulates the antioxidant response capacity of OSU and MEC-1 cells, the activity of glutathione reductase (GR) and glutathione peroxidase (GPx), two enzymes involved in glutathione-mediated antioxidant defenses, was evaluated. Data presented in Figure 9 show that, under basal conditions, the two cell lines exhibited similar activities for both enzymes (Figure 9A,B), which were inhibited only after SAS treatment (Appendix A, respectively). Interestingly, in the presence of a pro-oxidant stimulus as the H_2_O_2_ addition, both enzymatic activities increased by about 25%, confirming the adaptive response capability of both cell lines.

In addition, under oxidizing conditions, GR and GPx activities in OSU cells were unaffected by treatment with PRIMA-1^Met^, either alone or in combination with SAS, which, on the other hand, led to a decrease of approximately 25% in both activities. Conversely, in MEC-1 cells, the treatment with either PRIMA-1^Met^ or SAS alone caused a reduction in GR and GPx activities, although the reduction induced by SAS appeared more marked. However, the lowest activities of the two antioxidant enzymes were observed with the combination of treatments in a dose-dependent manner (Figure 9A and Figure 9B, respectively). The inhibition of antioxidant defenses was accompanied by an increase in NADPH oxidase activity, a pro-oxidant enzyme that, like GR, uses NADPH as a substrate. Specifically, NADPH oxidase activity increased slightly but significantly in OSU cells treated with PRIMA-1^Met^ and more markedly in the presence of SAS and the combination of treatments, whereas in MEC-1 cells, the increase in NADPH oxidase activity was already evident with PRIMA-1^Met^ alone and further increases in the presence of SAS and the combination (Figure 9C). As a consequence of the diminished activity of enzymatic antioxidant defenses and the enhanced NADPH oxidase activity, OSU and, especially, MEC-1 cells showed an increase in intracellular MDA levels, which reached a maximum when the highest PRIMA-1^Met^ concentration tested was combined with SAS (Figure 9D). Taken together, the treatments with PRIMA-1^Met^, SAS, and their combination indicated that MEC-1 cells are more sensitive than OSU cells with respect to antioxidant defense and oxidative stress generation.

### 2.8. PRIMA-1^Met^ and SAS Significantly Affect the Redox Status, Especially in TP53 Mutant CLL Primary Cells

To verify whether the changes in antioxidant defense and the oxidative damage accumulation observed in the OSU and MEC-1 cell lines also occurred in cells isolated from CLL patients clustered based on the presence of wt or mutant p53, the intracellular levels of GSH and the cellular antioxidant capacity, as well as the MDA content, were evaluated in the presence of CpG-ODN/IL-15. As shown in Figure 10, CLL cells carrying mutant p53 displayed a lower GSH content, irrespective of treatment, and significant differences were found compared to wt p53 carrying cells under every treatment condition (Figure 10). Treatment with PRIMA-1^Met^ or SAS resulted in a drastic decrease in GSH levels in both wild-type and mutant p53 cells (Figure 10A). However, the most pronounced decrease in GSH amount was observed when the two treatments were combined. Interestingly, no differences in total glutathione content (reduced + oxidized forms; GSH + GSSG) were observed between CLL cells expressing wt or mutant p53 under basal conditions. However, treatment with PRIMA-1^Met^ or SAS alone, and even more so their combination, caused a decrease in total glutathione levels (GSH + GSSG), which was more pronounced in CLL cells with mutant p53 compared to those expressing the wt protein (Figure 10B). The decrease in GSH was also associated with the reduction of antioxidant capacity in CLL cells, both wt and mutant p53 cells, in the presence of PRIMA-1^Met^ and SAS, reaching a minimum when the two treatments were combined (Figure 10B).

As observed in the cell lines, a defect in the antioxidant system allowed the cells to accumulate oxidative damage, as demonstrated by the elevated MDA levels in both single treatments and the combinations, reaching the highest levels in mutant p53 CLL cells treated with PRIMA-1^Met^ plus SAS (Figure 10D). Again, a significant difference between wt and mutant p53 cells in the MDA content was observed already in untreated cells and was maintained for all treatment conditions (Figure 10D).

Thus, results obtained from primary CLL cells closely resembled those from the two CLL cell lines, demonstrating an impairment of mutant p53-expressing cells to handle oxidative stress under basal as well as PRIMA-1^Met^-treated conditions.

## 3. Discussion

In CLL, *TP53* gene mutations represent a critical hallmark of the disease since its mutations may strongly affect the success of the treatment protocols. While *TP53* mutations are found with a low incidence at diagnosis, their incidence rises in refractory CLL, making the disruption of the *TP53* gene the most important predictive biomarker in CLL [8,9].

The p53 protein is a promising target for novel anticancer therapies. To date, several molecules have been tested in various tumors capable of reactivating the p53 protein function or eliminating mutated p53 proteins, thereby promoting tumor progression [35]. Among these, PRIMA-1^Met^ has reached the most advanced stages of experimentation, and in the last years, several clinical trials on hematological malignancies such as Multiple Myeloma and Acute Myeloid Leukemia have been conducted with PRIMA-1^Met^ in combination with standard treatments [36].

In addition to its ability to reactivate mutated p53 proteins, PRIMA-1^Met^ induces glutathione depletion, which leads to ROS overproduction [20,21]. Consequently, the combination of drugs aimed at targeting mutant p53 and factors involved in the antioxidant response may significantly affect tumor cell viability [25]. The cytotoxic potential of PRIMA-1^Met^ has been recently investigated in 62 clinical CLL samples characterized for *TP53* mutations and p53 protein levels [28]. A comparable viability decrease and apoptosis induction following PRIMA-1^Met^ were observed in mutant and wt p53 samples. Besides this study on PRIMA-1^Met^ effects in CLL, no investigations have been conducted to explore the activity of PRIMA-1^Met^ in combination with drugs involved in the modulation of the antioxidant response in CLL cells.

Here, we have explored how the combined treatment of PRIMA-1^Met^ and Sulfasalazine (SAS), a drug targeting the SLC7A11/xCT system, impacts CLL cells’ survival and their antioxidant response. These studies were conducted on two CLL cell lines, namely OSU and MEC-1, which carry a wt and a mutant p53 protein, respectively, and in several primary cells obtained from CLL patients.

The results of our investigation show several issues regarding PRIMA-1^Met^ and SAS activity in CLL.

Firstly, the PRIMA-1^Met^/SAS combination induced a significant decrease in OSU and MEC-1 cell survival in comparison to PRIMA-1^Met^ alone. Interestingly, the two drugs have a synergistic effect at the lowest concentrations of PRIMA-1^Met^ and SAS (Figure 3). In CLL patients’ cells, PRIMA-1^Met^ alone was toxic in a concentration and time-dependent manner, with no significant difference related to the p53 status, and in agreement with the results previously reported [28]. However, although the response of the CLL patients’ cells was heterogeneous in terms of Annexin/PI positivity (Appendix A), the PRIMA-1^Met^/SAS combination synergistically decreases cell survival (Appendix A). This is a desirable condition as it allows the use of the lowest effective concentrations of PRIMA-1^Met^ and SAS to achieve significant cytotoxicity on cancer cells while minimizing side effects. Notably, the PRIMA-1^Met^ doses used to treat CLL primary cells (up to 5 μM PRIMA-1^Met^) did not affect the viability of PBMC, where the IC_50_ was reached at considerably higher PRIMA-1^Met^ concentrations than in CLL cells (Appendix A). As the drug doses giving a synergistic effect are found to be the lowest ones, a negligible cytotoxic effect on non-cancerous cells can be envisaged. Thus, these results indicate that PRIMA-1^Met^ in combination with SAS may be considered a valid strategy to sensitize and kill wt and mutant p53 carrying CLL cells.

The heterogeneous sensitivity of CLL cells measured with Annexin/PI assays towards PRIMA-1^Met^ and SAS (Figure 4) can be ascribed to the expression of SLC7A11/xCT in different cell models. The expression of SLC7A11/xCT has been proposed to be used as a predictive biomarker of PRIMA-1^Met^ sensitivity [23,24]. Indeed, the pharmacological inhibition of SLC7A11/xCT by SAS could give rise to a synthetic lethal interaction with the mutant p53, resulting in high oxidative stress and cell death [25]. In this view, we show that MEC-1 cells express a lower level of SLC7A11/xCT compared to OSU, which could contribute to the higher PRIMA-1^Met^/SAS sensitivity observed. With these premises, we cannot rule out the possibility that a stratification of CLL patients’ samples based on SLC7A11/xCT expression could provide a clearer picture of PRIMA-1^Met^ sensitivity in combination with SAS, allowing for better customization of therapeutic interventions.

Secondly, PRIMA-1^Met^ was not able to transcriptionally reactivate the mutant p53 protein activity towards target genes. To identify a possible activity of PRIMA-1^Met^ on the transactivating potential of mutant p53, the modulation of p53 (wt and mutant), MDM2, and p21, following PRIMA-1^Met^ exposure, was assayed in OSU, MEC-1 (Figure 5), and primary CLL samples (Figure 6). In both experimental settings, we observed an increase in p53 level in cells expressing wt p53 that correlated with MDM2 and p21 levels (Figure 5A and Figure 6A). In contrast, in CLL primary cells carrying mutant p53, we observed a decrease in p53 as PRIMA-1^Met^ concentration increased, and no signals for p21 and MDM2 were detected (Figure 6, C). In MEC-1, after an increase in p53 at 5 μM PRIMA-1^Met^, we observed a dose-dependent reduction of mutant p53, MDM2, and p21 (Figure 5B), indicating the lack of mutant p53 reactivation by PRIMA-1^Met^. This is not surprising since the *TP53* gene in these cells presents a mutation at codon Q317, which gives rise to a frameshift mutation, generating a truncated form of the protein [37]. Thus, this mutant p53 protein would be unlikely to assume a wt-like conformation following PRIMA-1^Met^ treatment. In addition, this mutation is localized in proximity to the tetramerization domain of p53, making it difficult to form a transcriptionally active p53 tetramer. Based on these considerations, we can infer that in MEC-1 cells, the activity of PRIMA-1^Met^ could be dependent on its direct action on glutathione and the antioxidant-related system. It is interesting to note that the results obtained in cell lines resemble those obtained in CLL primary cells pre-treated with CpG-ODN/IL15, suggesting that OSU and MEC-1 cell lines can be considered a good model to study the response of activated, not quiescent, CLL cells to p53-targeting molecules. The modulation of p53 in primary CLL following PRIMA-1^Met^ in the presence of CpG-ODN/IL15 as an activator stimulus has not been investigated before. Herein, our data show that the pre-treatment with CpG-ODN/IL15 changes the modulation of mutant p53 and p21 at the lowest PRIMA-1^Met^ concentration, but, in any case, at higher drug concentrations, a decrease in mutant p53 and p21 was observed. The mutant p53 protein decrease following PRIMA-1^Met^ treatment has been previously reported and correlated to PRIMA-1^Met^ sensitivity in CLL samples [28]. Furthermore, the ability of PRIMA-1 to trigger the degradation, via ubiquitylation, of mutant, but not wt, p53 protein has been reported in breast cancer cells [38]. Thus, since the accumulation of a mutant p53 in cancer cells can be responsible for chemoresistance, it cannot be excluded that the observed cell sensitivity to PRIMA-1^Met^ in mutant p53 carrying cells is partially related to mutant p53 protein reduction.

Thirdly, CLL cells carrying mutant p53 have an impaired ability to manage oxidative stress and are more prone to accumulating ROS following PRIMA-1^Met^ and SAS treatment than wt p53-carrying cells. PRIMA-1^Met^ per se induced a decrease in cellular GSH, the active form of glutathione involved in protection against oxidative stress, which was more pronounced in MEC-1 than in OSU cells. GSH levels were reduced following treatment with PRIMA-1^Met^ alone and decreased further upon exposure to SAS; however, the most significant depletion was observed with the combination of the two compounds (Figure 8) and in MEC-1 cells than in OSU cells. As a consequence, ROS levels significantly increased in MEC-1 cells after treatment with PRIMA-1^Met^ and SAS, while only a modest increase was detected in OSU cells under the same conditions (Figure 8). These findings suggest that CLL cells with mutant p53 struggle with significant GSH depletion due to their impaired ability to counteract elevated ROS levels.

P53 can modulate glutathione metabolism by regulating genes involved in glutathione synthesis, recycling, and maintaining redox cofactors, thereby controlling cellular antioxidant capacity and oxidative stress responses [14]. On the other hand, it is important to note that reduced GSH levels may depend on several mechanisms, including (i) increased GSH consumption to buffer oxidative stress, (ii) impaired reduction of oxidized glutathione by GR, and (iii) limited availability of GSH precursors. Regarding the latter, our data show that both single and combined treatments with PRIMA-1^Met^ and SAS led to an upregulation of SLC7A11/xCT, more pronounced in MEC-1 than in OSU cells (Figure 7), indicating enhanced cystine uptake and potentially greater cysteine availability. The induction of SLC7A11/xCT by PRIMA-1^Met^ has previously been reported [23,32] and attributed to the inability of mutant p53 to bind Nrf2 and, in turn, inhibit xCT transcription. Also, the hypothesis of increased GSH consumption as a compensatory response to oxidative stress does not fully align with our results because GPx activity decreased in MEC-1 cells treated with PRIMA-1^Met^ or SAS and remained unchanged in OSU cells following PRIMA-1^Met^, although it declined after SAS exposure (Figure 9B). Instead, the drop in GSH levels may be explained by the reduced GR activity (Figure 9A), which follows the same trend observed for GPx. Furthermore, decreased GSH content may also result from enhanced NADPH oxidase activity in response to PRIMA-1^Met^, SAS, or their combination (Figure 9C). Since this pro-oxidant enzyme utilizes NADPH as a cofactor, it could limit the availability of this substrate for GR-mediated GSH production. Taken together, these alterations contribute to the oxidative damage accumulation, as evidenced by increased lipid peroxidation (Figure 9D), supporting the hypothesis that treatment with PRIMA-1^Met^, SAS, or their combination induces a redox imbalance, particularly in mutant p53-expressing cells.

A similar redox balance response to PRIMA-1^Met^ and SAS was observed in primary CLL samples (Figure 10). Notably, cells with mutant p53 exhibited impaired antioxidant defenses even under basal conditions. In all primary samples analyzed, PRIMA-1^Met^, SAS, and their combination led to a marked reduction in reduced GSH levels despite total glutathione content remaining unchanged. This suggests, as in the CLL cell lines, that the defect lies not in glutathione availability but in its conversion to the reduced form. Additionally, a general decline in antioxidant capacity was observed in both wt and mutant p53-expressing cells. This compromised defense further contributed to the accumulation of oxidative damage following treatment with PRIMA-1^Met^ or SAS, alone or in combination. Consistent with observations in cell lines, primary CLL cells carrying mutant p53 showed the highest levels of lipid peroxidation, particularly after the double treatment.

## 4. Conclusions

Taken together, our results show that in CLL cells, PRIMA-1^Met^ significantly influences the antioxidant response and the induction of ROS, decreasing cell survival. The combination of PRIMA-1^Met^ and SAS further causes a redox imbalance and increases oxidative stress in both wt and mutant p53 CLL cells, resulting in a synergistic effect on cell survival. However, CLL cells with mutant p53 are more severely compromised in their ability to handle oxidative stress compared to cells expressing wild-type p53. Future studies will focus on understanding how the metabolic pathway is impaired in mutant p53 CLL cells, with special attention to the modulation of glutathione-related proteins (such as SLC7A11). Additionally, examining how and if specific *TP53* mutations in CLL patients affect the response to PRIMA-1^Met^ and SAS will help us target this vulnerability to improve CLL therapies.

## 5. Materials and Methods

### 5.1. Cell Lines and Drug Treatments

OSU CLL cell line (TP53 wild-type) was obtained from the Ohio State University and previously described [39]. MEC-1 cell line (TP53 p.Q317fs/del17p) [29] was obtained from Deutsche Sammlung von Mikroorganismen und Zellkulturen GmbH (DSMZ, Braunschweig, Germany). The p53 status of these cells was confirmed by DNA sequencing. Cells were grown in RPMI containing 10% fetal bovine serum (FBS), L-glutamine, and penicillin-streptomycin antibiotic mixture (Euroclone, Milan, Italy), and maintained at 37 °C, in 5% CO_2_, at 100% humidity. PRIMA-1^Met^ (ab145974, Abcam, Cambridge, UK) was dissolved in H_2_O at 50 mM concentration, and Sulfasalazine (SAS, Cayman Chemical, Ann Arbor, MI, USA) was dissolved in DMSO at 50 mM concentration. Working solutions were prepared by appropriate dilutions in a serum-free medium. H_2_O_2_ (Farmac-Zabban S.p.A., Bologna, Italy) was diluted in PBS from the 880 mM stock solution and used at 50 mM final concentration. For the PRIMA-1^Met^ and SAS combined treatments, SAS was added 4 h before the addition of PRIMA-1^Met^. For treatment in oxidant conditions, H_2_O_2_ was added 2 h before SAS.

### 5.2. CLL Patients’ Cell Preparations and Treatment

CLL cells were obtained from the peripheral blood of CLL patients, after informed consent according to the Declaration of Helsinki. CLL patients were recruited in the Vivo-CLL protocol, a prospective monocentric study promoted in the IRCCS Ospedale Policlinico San Martino and approved by the Regional Ethics Committee. Appendix A summarizes the clinical features of the patients investigated. At the time of sampling, all CLL patients were untreated. Mononuclear cells were isolated by Ficoll (Lympholyte-H, Cedarlane Labs, distributed by Euroclone, Milan, Italy) density gradient centrifugation. PBMCs from patients with CLL were isolated by Ficoll-Hypaque (Lympholyte-H, Cedarlane Labs, distributed by Euroclone, Milan, Italy) density gradient centrifugation, and CD19-positive CLL cells were enriched by negative selection with the B-CLL isolation Kit (Miltenyi Biotec Srl, Bologna, Italy). The percentage of purified B cells (CD19+) exceeded 95%. Mononuclear cells were isolated by Ficoll (Lympholyte-H, Cedarlane Labs, distributed by Euroclone, Milan, Italy) density gradient centrifugation. Viable cell counts of CLL samples were conducted before each experiment by trypan blue staining and an automatic cell counter (TC20, Bio-Rad, Milan, Italy). Values >80% of live cells were considered suitable for the subsequent experimental procedures. CLL cells were cultured in RPMI culture medium supplemented with 10% FBS at high cell density (2–3 × 10^6^/^mL^). CLL cells activation was achieved by CpG/ODN2006 (hTLR9 ligand) (5 μg/mL) (Aurogene, Rome, Italy) + IL-15 (10 ng/mL) (PeproTech, Rocky Hill, NJ, USA). In vitro response to stimuli was confirmed flow cytometrically by increased cell size (forward light scattering). Activated and non-activated cells were treated with PRIMA-1^Met^ and SAS for 24 h, 48 h, and 72 h at the indicated concentrations, and apoptosis induction was assayed by Annexin V/PI staining and flow cytometry as detailed below.

### 5.3. MTT Assay

Cells were plated in 96-well plates at a density of 50 × 10^3^ cells per well in 100 μL of medium, cultured for 24 h, and treated with PRIMA-1^Met^ and or SAS for a further 24 h and 48 h. After treatments, 11 μL of 5 mg/mL Thiazol Blue Tetrazolium Blue (MTT, Sigma-Aldrich, St. Louis, MO, USA), dissolved in 10 mL of PBS, was added to each well, and plates were processed as described [40]. Absorbance was measured at 570 nm with a reference at 690 nm (Mithras LB 940, Multilabel reader, Berthold Technologies, Bad Wildbad, Germany).

### 5.4. Synergy Calculation

The isobole method was used to analyze the effects of combination treatments obtained with the MTT assay [41]. Following the indications, for a combined treatment between the drugs S (SAS) and P (PRIMA-1^Met^), the combination index D was calculated by the equation:D = Sc/Se + Pc/Pe
where Sc and Pc represented the drug concentrations used in combination, while Se and Pe were the drug concentrations of S and P that gave, alone, the same magnitude of effect [42]. For D = 1, the final effect was considered additive; if D < 1, the final effect of the combination was considered synergistic; if D > 1, the drug combination was considered antagonistic. Each combination was repeated 5 times. The Student’s *t*-test was used for statistical analysis. A mean D value for additivity (sham combination) was calculated utilizing combinations of two serial dilutions of the single drugs utilized in combinations.

In the case of CLL cells, considering that the dose–response curves for both drugs PRIMA-1^Met^ and SAS were linear, we applied the concept that the synergistic effect between the combination of the two drugs can be calculated using the “summation of doses” [41]. In particular, considering that if the two drugs act in an additive way, the expected combined effect will be given by the sum of the individual effects, i.e.,E_comb_ = E_a_ + E_b_.

In the case in which the effect is synergistic:E_comb_ > E_a_ + E_b_,
vice versa for the antagonistic effect.

In other words, if D_lin_ = E_comb_/E_a_ + E_b_ = 1, the result will be additive; if >1, it will be synergistic; if <1, it will be antagonistic.

### 5.5. Western Blotting Analysis

Protein extracts were prepared as previously described [43]. A 20 μg amount of cell lysates was resolved on 4–15% Mini Protean TGX precast gels (Bio-Rad, Milan, Italy) and transferred to nitrocellulose membranes by Trans-Blot Turbo Blotting System (Bio-Rad, Milan, Italy). Membranes were blocked with 2% non-fat dry milk in 0.1% Tween-20 in PBS for 1 h, then incubated 1 at room temperature or overnight at 4 °C with the appropriate primary antibody. The following antibodies were employed: anti-P53 (clone DO-1, Santa Cruz Biotechnology, Dallas, TX, USA), anti-MDM2 (clone SPM14, sc-965 Santa Cruz Biotechnology), anti-P21 Waf1/Cip1 (DCS60 #2946 Cell Signaling Technology, Danvers, MA, USA), anti-Bcl-2 (sc-509, Santa Cruz Biotechnology) anti-cystine/glutamate transporter (SLC7A11/xCT; Clone D2M7A, #12691 Cells Signaling Technology, Danvers, MA, USA), anti-human β-Actin (clone AC-74, Sigma–Aldrich, St. Louis, MO, USA). The appropriate IgG-horseradish peroxidase-conjugated secondary antibodies (anti-mouse or anti-rabbit IgG HRP, A9044 and A9169, respectively, Sigma-Aldrich, St. Louis, MO, USA) were used. Detection was carried out with ECL FAST PICO (ECL-1002, Immunological Sciences, Roma, Italy). Chemiluminescence was analyzed by Alliance LD, UVITEC Cambridge (Cambridge, UK).

### 5.6. Apoptosis Determination

OSU and MEC-1 apoptotic cells were analyzed using the Annexin V-FITC/PI double staining method. Cells were harvested 24 h after treatment, centrifuged for 5 min at 1000 rpm, and resuspended in binding buffer at a concentration of 2 × 10^5^ cells/mL according to the manufacturer’s instructions (Annexin V-FITC Apoptosis detection kit, Invitrogen, Milan, Italy). The percentage of cells undergoing early (FITC positive, PI negative) and late (FITC positive, PI positive) apoptosis was assessed by flow cytometry via dual-color analysis on 10,000 gated cells using a Cytoflex S flow cytometer (Beckman-Coulter, Brea, CA, USA). For the analysis of CLL primary cells, cells were harvested 24 h, 48 h, and 72 h after treatment and analyzed according to the protocol described above.

### 5.7. ROS Determination by Flow Cytometry

To evaluate the level of reactive oxygen species (ROS), cells were washed with PBS, resuspended in the same buffer, and stained for 20 min at 37 °C with 2′,7′-dichlorodihydrofluorescein diacetate (H_2_DCFDA, Thermo Fisher Scientific, Waltham, MA, USA) at a concentration of 5 μM in serum-free RPMI. The stock solutions were prepared in DMSO at 10 mM concentration [44]. H_2_DCFDA is a non-fluorescent dye, which is cleaved within the cells to 2′,7′-dichlorofluorescein (H_2_DCF). In the presence of oxidant agents, H_2_DFC is converted into the highly fluorescent 2′,7′-dichlorofluorescein (DCF). The analysis, performed with the Cytoflex S flow cytometer (Beckman Coulter, Brea, CA, USA), was conducted on 10.000 viable cells per condition. The probe was excited at 488 nm, and the emitted fluorescence was collected at 520 nm.

### 5.8. Determination of the Intracellular Level of GSH by HPLC

Intracellular levels of reduced glutathione (GSH) were assessed by high-performance liquid chromatography (HPLC) using the method reported by Fariss and Reed for total GSH [45]. Briefly, untreated and treated cells were harvested and centrifuged (300 rcf × 5 min). Then the pellets were washed in PBS, precipitated with 10% perchloric acid (PCA), and the thiol groups were blocked with iodoacetic acid (alkaline pH 8–9) for 10 min at room temperature in the dark. Next, the analytes were converted to 2,4-dinitrophenyl derivatives by incubating the samples with 1% 1-Fluoro-2,4-dinitrobenzene (FDNB) at 4 °C in the dark overnight. The quantitative determination of the derivatized analytes was carried out by HPLC equipped with an NH2 Spherisorb column and a UV detector set at 365 nm with a flow rate of 1.25 mL/min. The mobile phase A was maintained at 80% solution A (80% methanol in water) and 20% solution B (0.5 M sodium acetate in 64% methanol in water) for 5 min, followed by a 10 min linear gradient to 1% A and 99% B. Chromatography was performed with gradient elution. The data obtained were normalized to the intracellular amount of protein and expressed as μEq/mg of protein.

### 5.9. Evaluation of Antioxidant Enzymatic Activity

Glutathione reductase (GR) activity was evaluated spectrophotometrically following the NADPH oxidation at 340 nm. The mixture contained 100 mM Tris HCl, pH 7.4, 1 mM EDTA, 5 mM GSSH, and 0.2 mM NADPH [46].

Glutathione peroxidase (GPx) activity was assayed following the H_2_O_2_ decomposition at 240 nm, using an assay solution containing 100 mM Tris-HCl (pH 7.4), 5 mM H_2_O_2_, and 5 mM GSH. Since H_2_O_2_ is also a substrate of catalases, GPx activity is obtained by subtracting the result of this assay from the catalase activity values (catalase was assayed spectrophotometrically following the H_2_O_2_ decomposition at 240 nm) [46].

### 5.10. Evaluation of NADPH Oxidase Activity

NADPH oxidase activity was evaluated spectrophotometrically in the cell homogenates by measuring the reduction of cytochrome c at 550 nm using a solution containing 60 mM cytochrome c, 1 mM CaCl_2_, and 1 mM MgCl_2_. The assay was performed in the presence or absence (blank) of 20 mg/mL of superoxide dismutase (SOD). After incubation for 5 min at 37 °C, the sample was centrifuged at 800× *g* for 5 min, and the absorbance of the supernatant was evaluated at 550 nm. All activities were normalized to the protein concentration of the sample and expressed as mU/mg [47].

### 5.11. Evaluation of Malondialdehyde Content and Antioxidant Capacity

Malondialdehyde (MDA) concentration was assayed as a lipid peroxidation marker by the thiobarbituric acid reactive substances (TBARS) assay [48]. A 600 μL amount of TBARS solution was added to 50 μg of cell homogenate dissolved in 300 μL of Milli-Q water. The mix was incubated for 40 min at 100 °C. Afterward, the sample was centrifuged at 14,000 rpm for 2 min, and the supernatant was spectrophotometrically analyzed at 532 nm. Antioxidant capacity (AO), expressed as Trolox equivalent, and intracellular GSH content ratio in CLL primary cells were evaluated by commercial kits (Abcam; Cat #ab65329, and Merck; Cat #MAK440, respectively) following the manufacturer’s instructions.

### 5.12. Statistical Analyses

Unpaired *t*-test, ordinary one-way Anova, and two-way Anova, followed by multiple comparisons test, were performed using GraphPad Prism version 9.0. for Windows, GraphPad Software (San Diego, CA, USA). *p* < 0.05 was considered statistically significant.

## Figures and Tables

**Figure 1 ijms-26-05559-f001:**
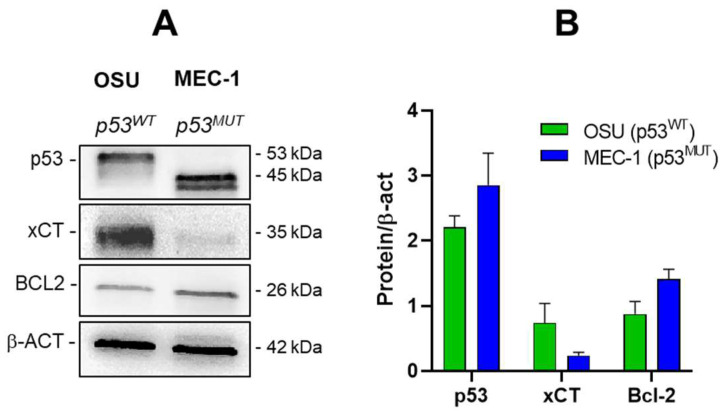
Protein levels of p53, SLC7A11/xCT, and Bcl-2 proteins in OSU and MEC-1 untreated cells. (**A**) Western blots are representative of three independent experiments (Appendix A). β-actin (β-ACT) is used for loading control; (**B**) Histogram showing the mean + SEM of p53, xCT, and Bcl-2 protein levels normalized for β-actin from three independent experiments (for original blot, see Appendix A).

**Figure 2 ijms-26-05559-f002:**
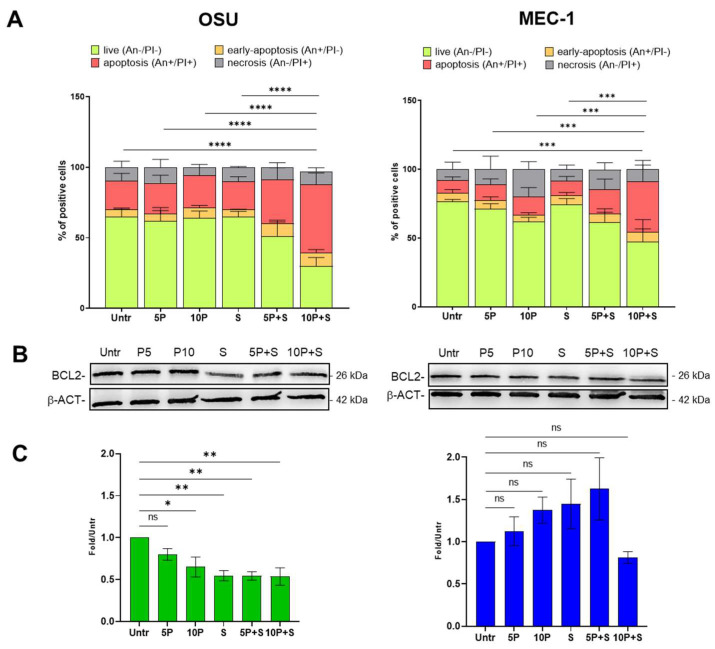
Apoptosis induction in OSU and MEC-1CLL cell lines. Annexin V-FITC/PI assay was performed in both cell lines treated with 5 μM and 10 μM PRIMA-1^Met^ (P) alone or in combination with 300 μM Sulfasalazine (S). (**A**) The histograms show the means + SD of the percentage of Annexin/PI double-positive cells detected by cytofluorimetric assay of three independent experiments. The complete statistical analysis is reported in Appendix A; (**B**) Representative Western blots showing the levels of Bcl-2 protein in OSU and MEC-1 cells treated for 24 h with 5 μM or 10 μM PRIMA-1^Met^ alone or in combination with 300 μM Sulfasalazine (S). β-actin (β-ACT) was used as loading control (for original blots, see Appendix A); (**C**) The histograms showed the means ± SEM of Bcl-2 chemiluminescence, normalized for β-actin, calculated as fold over the levels determined in untreated cells, of at least three independent experiments; ns = not significant, * *p* < 0.05, ** *p* < 0.01, *** *p* < 0.001, **** *p* < 0.0001.

**Figure 3 ijms-26-05559-f003:**
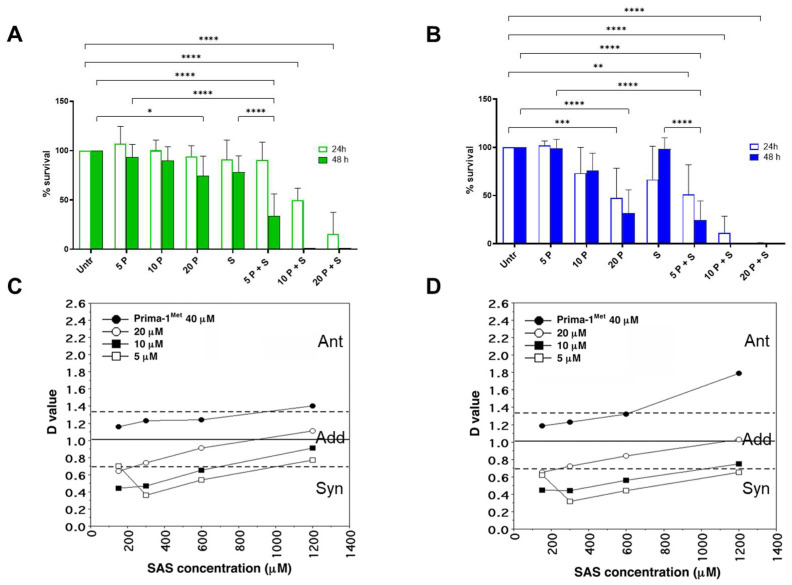
Cell viability of OSU and MEC-1 cells treated with PRIMA-1^Met^ alone and in combination with SAS. Cell viability after 24 h and 48 h of PRIMA-1^Met^ (P) and/or Sulfasalazine (S) treatments in OSU (**A**) and MEC-1 (**B**) cell lines. Cell viability was analyzed by MTT assays in cells exposed for 24 h and 48 h to increasing concentrations of PRIMA-1^Met^ (5, 10, 20 μM) alone or in combination with Sulfasalazine (300 μM). The histograms report the means ± SD of at least three independent experiments. D values, calculated after 48 h of treatment, were obtained for the combinations of PRIMA-1^Met^ and SAS on OSU (**C**) and MEC-1 cells (**D**). PRIMA-1^Met^ concentrations were: 40 (●), 20 (○), 10 (■), and 5 μM (□). SAS concentrations were: 1200, 600, 300, and 150 μM. The experimental D value for additivity was calculated using combinations of two serial dilutions of the tested compounds. Mean ± SD: 1.01 ± 0.32 (*n* = 36). * *p* < 0.05, ** *p* < 0.01, *** *p* < 0.001, **** *p* < 0.0001.

**Figure 4 ijms-26-05559-f004:**
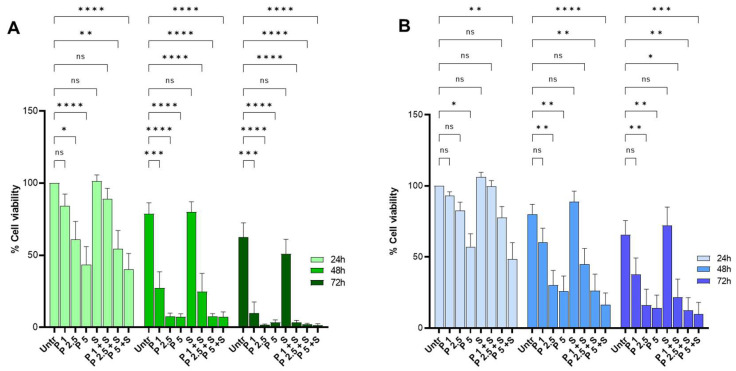
PRIMA-1^Met^ and SAS reduce the viability of wt and mutant p53 CLL cells. Wt p53 (**A**) and mutant p53 (**B**) carrying CLL cells were stimulated with CpG-ODN/IL15 and treated with PRIMA-1^Met^ (P) and Sulfasalazine (S) at the indicated concentrations (1, 2.5 and 5 μM PRIMA-1^Met^, 300 μM Sulfasalazine) for 24 h, 48 h, and 72 h and viability was calculated as the percentage of double Annexin V and PI negative cells. For each point, the percentage of viable cells was normalized for the percentage of viable cells in the corresponding untreated sample (100%). The histogram reports the average of the values (from 6 CLL patients carrying wt p53; from 8 CLL patients carrying mutant p53) + SEM; ns not significant; * *p* < 0.05, ** *p*< 0.01, *** *p* < 0.001, **** *p* < 0.0001.

**Figure 5 ijms-26-05559-f005:**
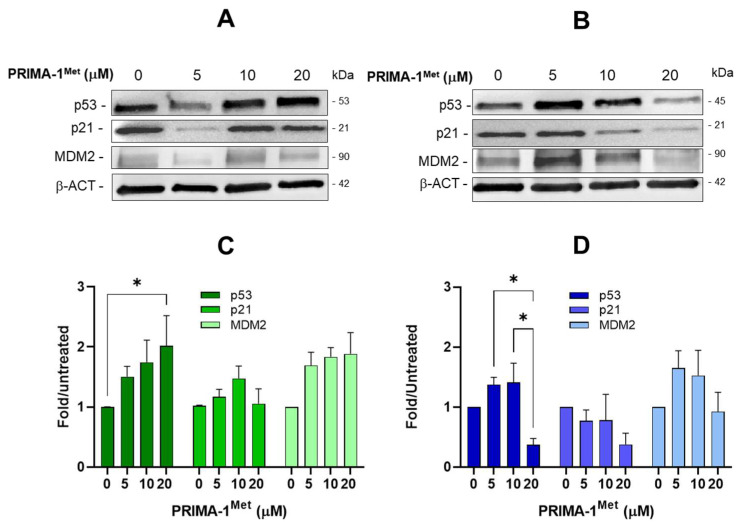
Modulation of p53, p21, and MDM2 in OSU and MEC-1 cells following PRIMA-1^Met^ treatments. Representative Western blots showing the expression of p53, p21, and MDM2 proteins in OSU (**A**) and MEC-1 (**B**) cells treated for 24 h with 5–20 μM PRIMA-1^Met^. β-actin (β-ACT) was used as a loading control. Histograms showing the expression of p53, p21, and MDM2 proteins, normalized for β-actin, in OSU (**C**) and MEC-1 (**D**) cells treated for 24 h with 5–20 μM PRIMA-1^Met^. Reported are the means ± SEM of chemiluminescence calculated as fold over the same protein levels in untreated cells of four independent experiments: * *p* < 0.05. (For original blots, see Appendix A).

**Figure 6 ijms-26-05559-f006:**
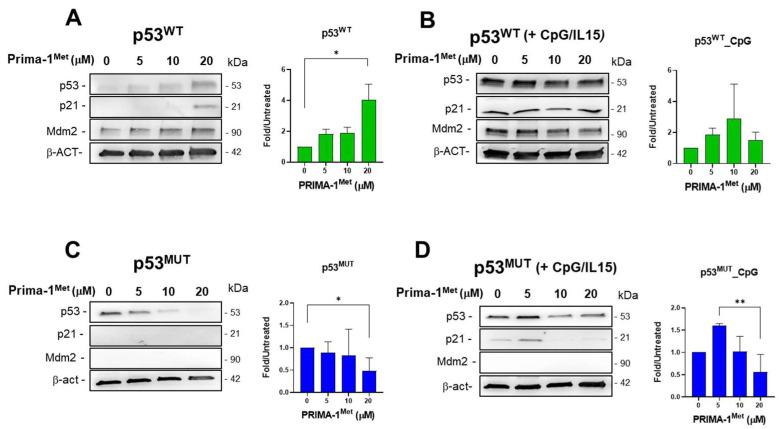
Modulation of p53 by PRIMA-1^Met^ treatment in the absence or the presence of CpG-ODN/IL15. (**A**) Representative Western blot of wt p53, MDM2, and p21 protein modulation after PRIMA-1^Met^ (sample #16, Appendix A); β-actin was used for loading control and normalization; the relative histogram on the right represents p53 data from three independent samples (samples #10, #16, #17, Appendix A) reported as means + SEM; (**B**) Representative Western blot of wt p53, MDM2 and p21 protein modulation after PRIMA-1^Met^ and CpG-ODN/IL15 pre-treatment (sample #16, Appendix A); the relative histogram on the right represents p53 data from two independent samples (samples #16, #17, Appendix A) reported as means + SEM; (**C**) Representative Western blot of mutant p53, MDM2 and p21 protein modulation after PRIMA-1^Met^ (sample #9, Appendix A); the relative histogram on the right represents p53 data from three independent samples (samples #3, #4, #9, Appendix A) reported as means + SEM; (**D**) Representative Western blot of mutant p53, MDM2 and p21 protein modulation after PRIMA-1^Met^ and CpG-ODN/IL15 pre-treatment (sample #9, Appendix A); the relative histogram on the right represents p53 data from two independent samples (samples #4, #9) reported as means + SEM; * *p* < 0.05, ** *p* < 0.01. (For original blots, see Appendix AA,B).

**Figure 7 ijms-26-05559-f007:**
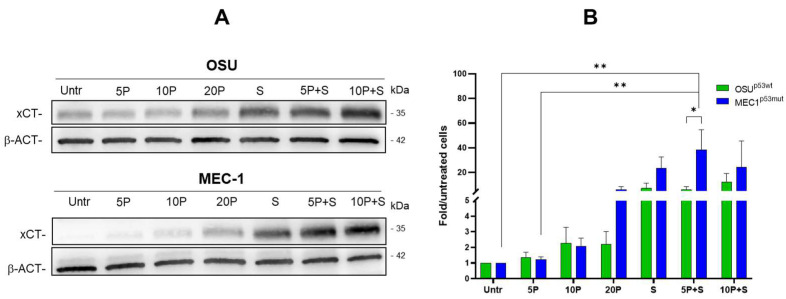
Levels of SLC7A11/xCT in OSU and MEC-1 cells. (**A**) Representative Western blot showing the levels of SLC7A11/xCT protein in OSU and MEC-1 cells treated with PRIMA-1^Met^ (5 µM, 10 µM, 20 µM) alone or in combination with 300 µM SAS after 24 h. β-actin (β-ACT) was used as a loading control; (**B**) Histogram showing the expression of SLC7A11/xCT protein in OSU and MEC-1 cells treated for 24 h with 5 μM–10 μM PRIMA-1^Met^ alone or in combination with 300 μM SAS. Reported are the means ± SEM of chemiluminescence calculated as fold over the levels of the same proteins in untreated cells of at least three independent experiments. β-actin is the internal control for normalization; * *p* < 0.05, ** *p* < 0.01. (For original blots, see Appendix A).

**Figure 8 ijms-26-05559-f008:**
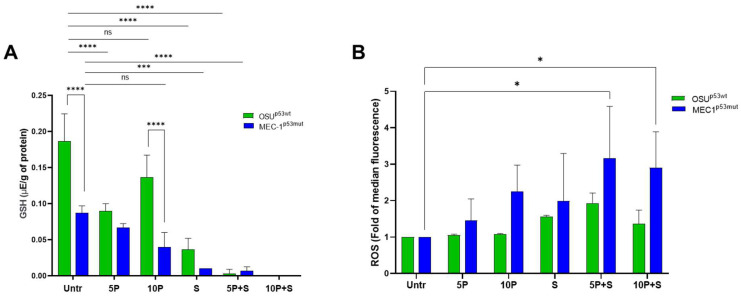
Levels of GSH and ROS in OSU and MEC-1 cells following PRIMA-1^Met^ and SAS treatments. (**A**) The levels of GSH were measured by HPLC analysis in OSU and MEC-1 cells treated with PRIMA-1^Met^ (5 µM or 10 µM) alone or combined with 300 µM SAS after 24 h. The histogram reports the mean ± SD of three independent experiments. GSH concentrations were expressed as µE/g of total protein; (**B**) Levels of ROS in OSU and MEC-1 cells treated with PRIMA-1^Met^ alone or in combination with SAS after 24 h. Data are reported as mean of fluorescence ± SD obtained with cytofluorimetric analysis and calculated as fold over the levels observed in untreated cells in three independent experiments; ns not significant, * *p* < 0.05, *** *p* < 0.001, **** *p* < 0.0001.

**Figure 9 ijms-26-05559-f009:**
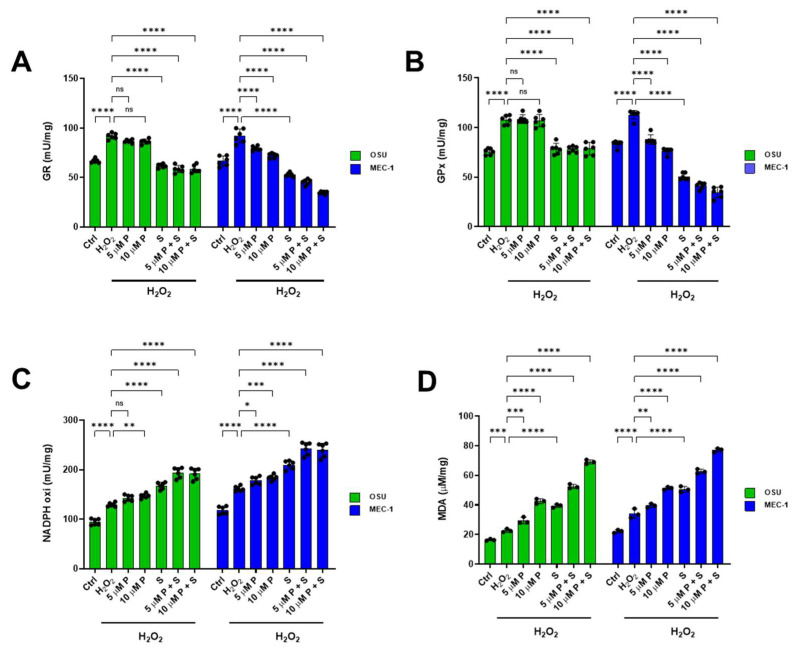
Enzymatic activity involved in redox balance and lipid peroxidation accumulation in OSU and MEC-1 cells pre-treated with H_2_O_2_. Determination of (**A**) glutathione reductase (GR) activity, (**B**) glutathione peroxidase (GPx) activity, (**C**) NADPH oxidase activity, and (**D**) malondialdehyde (MDA, a lipid peroxidation marker) intracellular level in H_2_O_2_ pre-treated OSU and MEC-1 cells following PRIMA-1^Met^ alone and in combination with SAS. Each panel is representative of three independent experiments; data are reported as means + SD; ns, not significant, * *p* < 0.05; ** *p* < 0.01, *** *p* < 0.001, **** *p* < 0.0001.

**Figure 10 ijms-26-05559-f010:**
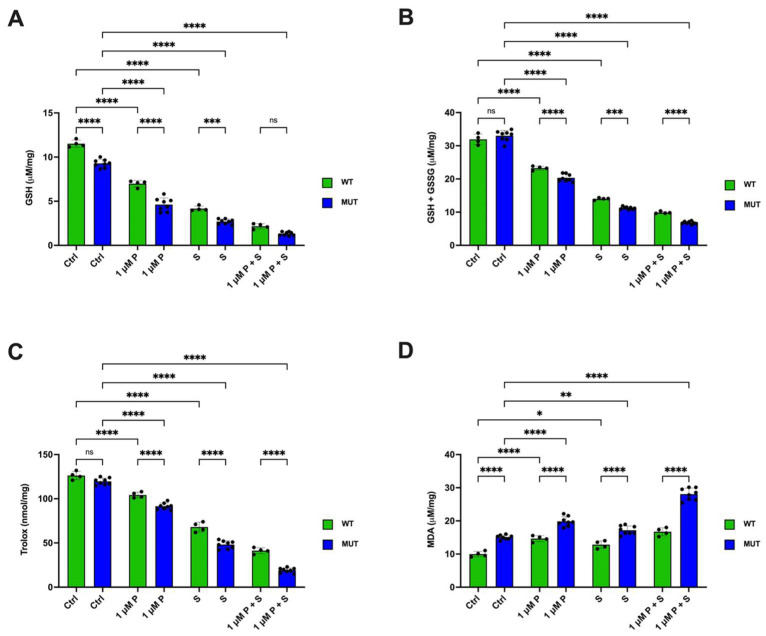
Metabolic markers in CLL patients’ cells. Determination of (**A**) GSH level, (**B**) GSH + GSSG content, (**C**) cellular total antioxidant capacity, (**D**) malondialdehyde (MDA) intracellular level in CLL patients’ cells following 1 μM PRIMA-1^Met^ alone and in combination with 300 μM SAS. Each panel is representative of three independent experiments; data are reported as means + SD. Green columns, wt p53 CLL cells; blue columns, mutant p53 CLL cells. ns, not significant, * *p* < 0.05; ** *p* < 0.01, *** *p* < 0.001, **** *p* < 0.0001.

## Data Availability

The data sets generated and/or analyzed during the current study are available from the corresponding author upon reasonable request.

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
