# Peer review of "Targeting the p53/xCT/GSH Axis with PRIMA-1Met Combined with Sulfasalazine Shows Therapeutic Potential in Chronic Lymphocytic Leukemia"

_ijms, 2025, doi:10.3390/ijms26125559_

Round 1

Reviewer 1 Report

Comments and Suggestions for Authors

Targeting the p53/xCT/GSH axis with PRIMA-1Met combined 2 with Sulfasalazine shows therapeutic potential in chronic lym- 3 phocytic leukemia

  1. This sentence is part of abstract, for me long. Could you please be kind to make shorter? Because for reader must be easy to understand. In this study, we investigated whether the association of PRIMA-1Met with Sulfasalazine (SAS), a molecule that inhibits the cystine/gluta- 36 mate transporter SLC7A11/xCT, leads to a decrease in CLL cell viability by targeting both 37 mutant p53 and the glutathione pathway.
  2. The same please make it shorter because for reader will be easy to understand. The combined PRIMA-1Met and SAS treatment synergistically decreased cell survival regardless of the p53 status and further impaired the antioxidant capacity, especially in cells with mutant p53, indicating that cell death induced by PRIMA-1Met and SAS is related to redox balance.
  3. Which type of mutation in P53 (in cell) you have exanimated? Based on my knowledge P53 have many types of mutation.
  4. How P53 regulate Gluthatione metabolism?
  5. You have mention ibrutinib and idelalisib for regulation of signaling pathways of apoptosis and P53 in CLL. What about other drugs such as Duvelisib, Venetoclax, Plerixafor, Dasatinib, GSK3β Inhibitors.

Reviewer 2 Report

Comments and Suggestions for Authors

This is an interesting study evaluating the therapeutic potential of the combination of Eprenetapopt and sulfasalazine in chronic lymphocytic leukemia, in particular if the combination target both mutant p53 and the glutathione pathway in treating CLL. 

The details included in the paper are well summarized in the figures, but can be improved by an inclusion of a diagram at the beginning of the paper detailing where eprenetapopt fits in the pathogenesis of CLL - where it is as a potential treatment target, can improve readers' understanding, especially how its combination with sulfasalazine may work. Similarly, a schematic diagram showing the role of glutathione and GSH in diseases, will also be a useful addition for the readers' understanding of the rationale of this combination.

Results on the synergism of both compounds (Figures 3c and d) were slightly confusing with the description of the variable activity in the cell lines - antagonistic activity at the higher concentration of eprenetapopt- can this be clarified as the authors report that altogether, the data indicated that combination synergistically influence CLL cell survival- is this concentration dependent?

The description of results in CLL patient samples- are these samples at diagnosis or relapsed? If samples are at different timepoints - do the results vary with disease relapse or progression?

Discussion of the results were detailed, but the conclusions can benefit from perhaps some suggestions of strategies to the next step to advance their findings.
